# Correlation of Diffusion Tensor Tractography with Restless Legs Syndrome Severity

**DOI:** 10.3390/brainsci13111560

**Published:** 2023-11-07

**Authors:** Kang Min Park, Keun Tae Kim, Dong Ah Lee, Yong Won Cho

**Affiliations:** 1Department of Neurology, Haeundae Paik Hospital, Inje University College of Medicine, Busan 48108, Republic of Korea; smilepkm@hanmail.net (K.M.P.); h00533@paik.ac.kr (D.A.L.); 2Department of Neurology, Keimyung University School of Medicine, Daegu 42601, Republic of Korea; 6k5upa@gmail.com

**Keywords:** restless legs syndrome, diffusion tensor imaging, white matter

## Abstract

This prospective study investigated white matter tracts associated with restless legs syndrome (RLS) severity in 69 patients with primary RLS using correlational tractography based on diffusion tensor imaging. Fractional anisotropy (FA) and quantitative anisotropy (QA) were analyzed separately to understand white matter abnormalities in RLS patients. Connectometry analysis revealed positive correlations between RLS severity and FA values in various white matter tracts, including the left and right cerebellum, corpus callosum forceps minor and major, corpus callosum body, right cingulum, and frontoparietal tract. In addition, connectometry analysis revealed that the FA of the middle cerebellar peduncle, left inferior longitudinal fasciculus, left corticospinal tract, corpus callosum forceps minor, right cerebellum, left frontal aslant tract, left dentatorubrothalamic tract, right inferior longitudinal fasciculus, left corticostriatal tract superior, and left cingulum parahippocampoparietal tract was negatively correlated with RLS severity in patients with RLS. However, there were no significant correlations between QA values and RLS severity. It is implied that RLS symptoms may be potentially reversible with appropriate treatment. This study highlights the importance of considering white matter alterations in understanding the pathophysiology of RLS and in developing effective treatment strategies.

## 1. Introduction

Restless legs syndrome (RLS) is a neurological disease that causes discomfort in the legs while resting, mainly in the evening [1]. RLS is a common disease that has a significant impact on quality of life, but the pathophysiology of RLS has not yet been clearly elucidated [2,3]. The documented prevalence of RLS ranges from 3.9% to 14.3%, depending on the specific population under investigation and the criteria applied [2,3]. Even with the widespread occurrence and significant impact of RLS, the underlying mechanisms remain incompletely comprehended. It is primarily known to be caused by neurotransmitter abnormalities or iron deficiency, but in recent studies, its relationship with brain abnormalities such as brain network changes has garnered increasing attention [4,5]. In patients with RLS, several functional brain networks, such as the thalamic, salience, default-mode, and small-world networks, have been known to undergo changes compared to healthy controls [4]. A study that performed structural connectivity analysis also showed that the brain network was different from that of the healthy control group, and this could be related to RLS symptoms [5].

The brain’s connectome serves as a map outlining the cortical connections between different regions [6,7]. Diffusion magnetic resonance imaging (MRI) has become the modality most widely utilized to measure the structural connectome in humans [7]. This technique employs a fiber tracking algorithm to map macroscopic connections from one end to another between parcellated gray matter [8,9]. While diffusion MRI-based tractography has garnered increasing popularity over the last decade, recent studies have highlighted concerns regarding the precision of measuring end-to-end connectivity. These fiber tracking algorithms have shown limited reliability, particularly near gray matter targets, raising doubts about the efficacy of these “find-difference-in-track” techniques [10,11]. To address the limitations of end-to-end fiber tracking, a novel concept, known as the local connectome, was introduced. The local connectome measures the degree of connectivity between adjacent voxels within a white matter fascicle, as determined by spin density. Understanding the local orientation and integrity of fiber bundles as they traverse the core of the white matter is essential for identifying where a bundle begins and ends. Thus, the local connectome serves as a fundamental unit of the end-to-end structural connectome and can function as a proxy for global connectivity analysis. This approach is known as connectometry [12,13]. Notably, a recent study utilized connectometry to investigate neuronal injuries in patients with mild traumatic brain injury. The correlation tractography revealed a connection between the splenium of the corpus callosum, the right superior longitudinal fasciculus, and the right cingulum, and performance on the digital symbol substitution test in patients with mild traumatic brain injury. This approach holds promise for shedding light on the intricacies of brain connectivity and its relationship with various neurological conditions [14]. However, no studies using this connectometry in patients with RLS have yet been conducted.

A diffusion tensor can yield several diffusivity measures, including fractional anisotropy (FA), axial diffusivity (AD), radial diffusivity (RD), and mean diffusivity (MD) [15]. FA measures the degree of anisotropy in the diffusion of water within tissues and is especially valuable for assessing the structural integrity of neural fibers in the brain. AD represents the rate of water diffusion along the axonal fibers, and RD indicates the rate at which water diffuses across the axonal bundles. The MD represents the average of the three eigenvalues of the tensor diffusivity [15,16]. Conversely, generalized q-sampling imaging (GQI) provides quantitative anisotropy (QA) and an isotropic diffusion component derived from GQI analysis (ISO), both based on diffusion density [17]. QA is computed using peak orientations on a spin distribution function (SDF), where each orientation defines a specific QA value. Notably, QA is defined per fiber orientation, whereas FA is defined per voxel. This distinction has a considerable impact on fiber tracking, as QA has proven valuable in filtering out false fibers when dealing with scenarios involving crossing fibers [17,18]. It is crucial to understand that the two types of measures, FA and QA, one based on diffusivity and the other on density, have distinct clinical meanings. However, only a few studies have focused on the differences of the FA and QA in patients with RLS.

This study aimed to investigate the tracts that correlated with RLS severity using correlational tractography in patients with RLS. In particular, we analyzed FA and QA separately to reveal the mechanism of the white matter abnormalities observed in these patients. We expected that the FA or QA values of specific white matter tracts would be highly correlated with RLS severity, and this tract may be related to the pathophysiology of RLS.

## 2. Materials and Methods

### 2.1. Participants

This prospective study was conducted at a single tertiary hospital. We asked patients with newly diagnosed RLS who visited our hospital whether they would like to participate in this study and enrolled them when they agreed. All the participants provided informed consent to participate in this study, which was approved by our institutional review board. Primary RLS was diagnosed by a sleep specialist based on the International RLS/WED Study Group (IRLSSG) criteria [1], as follows: (1) a compelling need to shift one’s legs, often accompanied by uncomfortable and disagreeable sensations in the legs, though not always; (2) the desire to move the legs, with associated discomfort arising or intensifying during moments of rest or inactivity, such as when lying down or sitting; (3) the impulse to move the legs, with the associated discomfort somewhat or completely alleviated through physical activity, like walking or stretching, as long as the activity persists; (4) the urge to move the legs, with the related discomfort during periods of rest or inactivity primarily manifesting or worsening during the evening or nighttime compared to the daytime; and (5) the presence of the above symptoms is not solely attributable to another medical or behavioral condition. The patients had no medical or neurological disease other than RLS. No structural brain MRI abnormalities were observed in the patients with RLS. Figure 1 shows the selection process of participants for this study.

Questionnaires, including the Insomnia Severity Index (ISI) [19], Pittsburgh Sleep Quality Index (PSQI) [20], and Hospital Anxiety and Depression Scale (HAS and HDS) [21], were administered to all study participants. They additionally completed the following two questionnaires: the International RLS Scale to assess the severity of RLS symptoms [22] and the Restless Legs Syndrome Quality of Life Questionnaire [23].

### 2.2. Diffusion Tensor Imaging MRI Acquisition

Patients with RLS underwent diffusion tensor imaging (DTI) using a 3.0T MRI scanner equipped with a 32-channel head coil (AchievaTx, Phillips Healthcare, Best, The Netherlands), which has been used for research purposes in patients with RLS. The specific DTI parameters were as follows: 32 different diffusion directions, b-values of 0 and 1000 s/mm^2^ (b0 images were acquired once), repetition time/echo time = 8620/85 ms, flip angle = 90°, slice thickness = 2.25 mm, acquisition matrix = 120 × 120, field of view = 240 × 240 mm^2^, and parallel imaging factor (SENSE) = 2. The phase direction was set in the anterior–posterior direction, and the fat was shifted posteriorly. T2-weighted images were also acquired to rule out structural lesions in the brain, and three-dimensional T1-weighted images (T1W) were also acquired because they were necessary in the preprocessing process of DTI. The T1W images were obtained using a turbo-field echo sequence with the following parameters: TI = 1300 ms, repetition time/echo time = 8.6/3.96 ms, flip angle = 8°, and 1 mm^3^ isotropic voxel size.

### 2.3. Connectometry Analysis with Statistical Analysis

A total of 69 diffusion MRI scans of patients with RLS were included in the connectometry database. We used the DSI studio program (version 2022 May, http://dsi-studio.labsolver.org, accessed on 8 July 2023) for preprocessing brain MRI, which included open-source images, correcting the eddy current and phase distortion artifacts, and setting up a mask (thresholding, smoothing, and defragmentation). The eddy current distortion is addressed through a Gaussian process for predicting the anticipated image, subsequently determining its conversion to match the real collected image. The correction involves reversing the distortion while also managing concurrent adjustments for rotation and translation caused by head movement. We also included a quality control step. Quality control plays a pivotal role in the analysis of diffusion MRI, since the contrast in diffusion MRI arises from signal attenuation caused by diffusion. Issues related to data quality, leading to signal loss, can simulate diffusion signal attenuation and, in turn, produce misleading tractography outcomes. Even with the correction methods discussed earlier, certain data may remain too corrupted to be effectively rectified due to various acquisition problems. Consequently, not all scan data are fit for processing. Therefore, diffusion MRI quality control should incorporate a screening phase for detecting and excluding data that cannot be adequately corrected. The diffusion data were reconstructed using the DTI method and GQI with a diffusion sampling length ratio of 1.25 [17]. GQI stands as a model-free technique employing fiber-resolving methods to gauge the orientations of fibers at each voxel in the imaging data. In model-free approaches, the goal is to determine orientation distribution functions, which can be likened to histograms that represent the observed diffusion distributions at various orientations. The FA values, which are DTI-based metrics, were used in the connectometry analysis. The QA, which is a GQI-based metric, was extracted as the local connectome fingerprint and also used in the connectometry analysis [24]. DTI connectometry was used to derive the correlational tractography that had a change in FA and QA correlated with RLS severity [12]. The connectometry follows a track-difference paradigm. Instead of mapping the entire end-to-end connectome, it focuses only on the segment of the fiber bundle, which demonstrates significant associations with the variables under study. It involves reconstructing the diffusion of MRI data into a standard template space to map the local connectome matrix for a group of subjects. Relevant study variables are then associated with this local connectome matrix to identify connections with significant correlations [12]. These localized connectomes are traced along the core pathway of the fiber bundle using a fiber tracking algorithm and then compared to the null distribution of coherent associations through permutation statistics [13]. A nonparametric Spearman partial correlation was used to derive the correlation, and the effect of sex and age was removed using a multiple regression model. A T-score threshold of 2.5 was assigned and tracked using a deterministic fiber tracking algorithm to obtain correlational tractography [24]. A seeding region was placed in the whole brain. We used the ICBM152 adult template to discover white matter tracks. The tracks were filtered by topology-informed pruning with four iterations [25]. A false discovery rate (FDR) threshold of 0.05 was used to select tracks to conduct multiple corrections. A total of 4000 randomized permutations were applied to the group labels to obtain a null distribution of the track length.

## 3. Results

### 3.1. Participants

Table 1 shows the demographic and clinical characteristics of the patients with RLS. Sixty-nine patients with RLS were enrolled. The mean age of the patients was 57 years. RLS was more common in women than in men, accounting for more than two-thirds of cases. The median age of onset was 47 years old.

### 3.2. Tracks with FA Correlated with RLS Severity

Table 2 summarizes the results of this study. The connectometry analysis revealed that the FA of the left cerebellum, right cerebellum, corpus callosum forceps minor, corpus callosum forceps major, corpus callosum body, and right cingulum frontoparietal track was positively correlated with RLS severity (FDR = 0.01) (Figure 2). In addition, connectometry analysis revealed that the FA of the middle cerebellar peduncle, left inferior longitudinal fasciculus, left corticospinal tract, corpus callosum forceps minor, right cerebellum, left frontal aslant tract, left dentatorubrothalamic tract, right inferior longitudinal fasciculus, left corticostriatal tract superior, and left cingulum parahippocampoparietal tract was negatively correlated with RLS severity in patients with RLS (FDR = 0.04) (Figure 3).

### 3.3. Tracks with QA Correlated with RLS Severity

The connectometry analysis found no significant result in tracks with QA positively or negatively correlated with RLS severity (FDR ≤ 0.05) (Figure 4 and Figure 5).

## 4. Discussion

This was the first study using correlational tractography targeting patients with RLS. Research has demonstrated that correlational tractography has the potential to unveil the structural mechanisms underpinning both brain function and dysfunction. These emerging methods call for further investigation to assess their clinical utility as innovative imaging biomarkers based on tracking [14]. In addition, this was also the first study to examine the relationship between RLS severity and QA. There were no significant results in white matter tracks with QA that were positively or negatively correlated with RLS severity. However, several white matter tracts with FA showed negative or positive correlations with RLS severity. Through this study, we were able to find white matter that was associated with the symptoms of RLS.

There was a difference between FA and QA. Myelination of axons inhibits diffusion; however, the DTI model cannot account for this effect. Due to the complexity of real-world biological changes, the use of DTI metrics results in substantial variation [26]. Due to Brownian motion, unrestricted and restricted diffusion are always combined in tensor-derived measurements of diffusivity, which is the greatest drawback. Separating them requires complex model fitting and typically does not work well for data with a low signal-to-noise ratio [26]. Meanwhile, the length parameter of the GQI specifies the distance scale for evaluating restricted diffusion. QA relies on q-space imaging to determine the densities of restricted and less-restricted diffusion. The distinction between restricted and less-restricted diffusion can be easily achieved through a straightforward linear relationship. Its performance was superior to that of DTI under conditions of a low signal-to-noise ratio. QA measures the density of diffusing water; consequently, QA measures the density of anisotropic diffusing water [27]. QA is also able to quantify the anisotropy of restricted diffusion and is therefore more resistant to inflammation and edema. A previous neurosurgical study demonstrated that QA was resistant to peritumoral edema and contributed to more trustworthy tractography [27]. Furthermore, QA quantified anisotropy for each population of fibers, whereas FA was shared by all populations of fibers within a voxel. QA is a fiber-specific measurement that is defined per fiber population and therefore provides a measurement for each individual fiber. The FA is defined per voxel, and all fiber populations within a voxel share the same measurement. FA can decrease in the presence of crossing fibers, whereas QA is less affected. QA is also less sensitive to the partial volume effects of crossing fibers, with better resolution [28,29]. Considering these differences between FA and QA, brain regions with decreased FA while QA remains unchanged could represent tissue edema but not necessarily a structural alteration in the axons. Neuronal changes such as demyelination, axonal loss, and/or edema reduce both FA and QA [17,18,27,30]. Therefore, brain regions in the present study showed a significant correlation between RLS severity and FA, but not QA, which suggests that the changes in white matter observed in patients with RLS may be reversible and not structural injury of the white matter tract. Therefore, RLS symptoms can be reversibly improved with appropriate RLS treatment. This suggests that active treatment of RLS is necessary.

Tracts that had a significant negative correlation with RLS severity were likely to be strongly associated with RLS. Because this study used a cross-sectional design, it is unknown whether RLS severity increased because of tract abnormalities or whether tract abnormalities occurred as RLS became more severe. However, the association between these tracts and RLS was clear and statistically significant despite multiple corrections. We found that the FA of the cerebellum, corpus callosum, and cingulum frontoparietal tract was positively correlated with RLS severity. A previous study showed that the cerebellum is related to RLS pathophysiology, which is consistent with our findings. A functional MRI study found an abnormal cerebellum–basal ganglia–sensorimotor cortex circuit in patients with RLS, underscoring the pivotal role of the cerebellum in RLS [31]. This suggests that RLS might have indirect involvement due to complex brain network interactions [31]. This may suggest that the cerebellum plays a significant role in motor control and the execution of movement, which could be relevant to RLS symptoms. In addition, patients with RLS often experience temporary relief from their RLS symptoms through activities such as movement or walking, which could be associated with the cerebellum’s role in motor control in patients with RLS [31]. The corpus callosum is a substantial bundle of nerve fibers located in the brain that connects the two cerebral hemispheres, which are the left and right sides of the brain. It is the largest white matter structure in the human brain and plays a crucial role in facilitating communication and coordination between the two hemispheres. The function of the corpus callosum includes interhemispheric communication, integration of sensory information, coordinated motor function, and language and cognitive functions [32,33]. A previous study investigated the corpus callosum shape differences between patients with RLS and healthy controls using a geometric morphometric approach and demonstrated callosal shape alterations in patients with RLS [32]. Furthermore, a previous DTI study in patients with RLS also showed decreased FA of the corpus callosum in patients with RLS [33]. The corpus callosum is the most important white matter area; it is composed of highly myelinated fibers that interconnect the two cerebral hemispheres. Myelin deficit is one of the hypotheses suggested in the etiology of RLS [32]. This research supports our finding that RLS severity is negatively correlated with the FA of the corpus callosum. Patients with RLS experience uncomfortable leg movements and sensations, and the frontal lobe may control or regulate these movements. Symptoms of RLS are sometimes related to sleep disturbances, and dysfunction of the frontal lobe can affect these symptoms. Furthermore, RLS symptoms are often associated with strong urges that make it difficult to suppress leg movements, and frontal lobe dysfunction may contribute to these inhibitory issues [33]. These reports support the results of the present study, that the white matter tracts associated with RLS severity are the cerebellum and corpus callosum. We can assume that FA values increased as a compensatory mechanism for RLS symptoms or as an indication of the hyperconnectivity of white matter tracts in patients with RLS. It can be hypothesized that as RLS symptoms become more severe, the connection of these white matter tracts increases, resulting in a compensation process to alleviate RLS symptoms; however, further research is needed to confirm this.

In addition, we found that FA in the middle cerebellar peduncle, inferior longitudinal fasciculus, corticospinal tract, corpus callosum, cerebellum, frontal aslant tract, dentatorubrothalamic tract, and cingulum parahippocampal-parietal tract were negatively correlated with RLS severity. It is not clear why the FA values of these structures were negatively correlated with RLS severity. It is unknown whether RLS severity increases as the FA value of these white matter tracts decreases, or whether FA values of these white matter tracts decrease as RLS severity increases. Although the cause and effect were unknown due to the cross-sectional study design, it is clear that these white matter tracts are related to RLS symptoms. In particular, a previous study that investigated white matter abnormalities in patients with RLS using DTI and tract-specific statistical analysis also proved that the FA value was lower than that of the normal group in the corticospinal tract, which is a major neural tract known to convey sensorimotor information [34]. Given that sensorimotor symptoms are the primary indicators of RLS, it is natural to consider that a decrease in FA within the corticospinal tract could potentially trigger abnormal transmission of sensorimotor signals [34]. A recent study using atlas-based analyses combining crossing-fiber-based metrics and tensor-based metrics based on the DTI demonstrated alterations in white matter tracts of the corticospinal tract, hippocampal cingulum, and inferior longitudinal fasiculus in patients with RLS; these results are consistent with our present study [35]. The white matter fiber pathways we found in our study were predominantly situated within the sensorimotor network. This aligns with the fiber pathways potentially linked to the involuntary compulsion to move one’s legs, a clinical hallmark of RLS. These discoveries offer substantiation of potential alterations in the central nervous system’s excitability at a pathophysiological level [35].

This study had some limitations. First, because this was a single-center study conducted at a single tertiary hospital, it is difficult to apply the results to general patients with RLS. However, the strength of this study is that we prospectively enrolled a large number of patients with RLS. Second, the cross-sectional design made it difficult to clarify the causes and effects of the results. A follow-up study measuring changes in QA and FA values of the white matter tract according to RLS treatment could prove our various hypotheses. Further longitudinal follow-up studies are required to overcome these limitations. In particular, before-and-after comparisons based on treatment for RLS in the same patients would be useful. Third, we performed multiple correlations and derived meaningful results but used an FDR method rather than a strict method such as the Bonferroni correction. The FDR is a way to balance the trade-off between identifying true discoveries and minimizing the number of false discoveries. We judged that multiple corrections using FDR was sufficient in this study. Fourth, we could not enroll healthy controls. Thus, it was not possible to find white matter tracts that correlated with RLS severity in the normal group and compare them with patients with RLS. Fifth, we used deterministic tractography. However, deterministic tractography follows a single, most likely pathway for each voxel, while probabilistic tractography considers the uncertainty in diffusion measurements and models multiple possible pathways, providing more robust and accurate results in complex brain regions [36]. Thus, recent studies have often used probabilistic tractography rather than deterministic tractography [37]. Lastly, the evaluation in this study solely focused on white matter changes in patients with RLS. Future studies should consider utilizing multiple modalities, such as assessing cortical thickness, cortical surface area, and cortical volume for the cerebral cortex, investigating dynamic changes in functional networks through resting state functional connectivity analysis, and examining cerebral blood flow using arterial spin labeling technology. White matter and gray matter and structural and functional connectivity do not exist independently but are related to each other, so they must be comprehensively judged using various modalities.

## 5. Conclusions

We found that the FA of several white matter tracts was positively or negatively correlated with RLS severity, but this was not the case for QA. This suggests that the changes in white matter observed in patients with RLS could be reversibly improved with appropriate RLS treatment. This suggests that active treatment of RLS is necessary.

## Figures and Tables

**Figure 1 brainsci-13-01560-f001:**
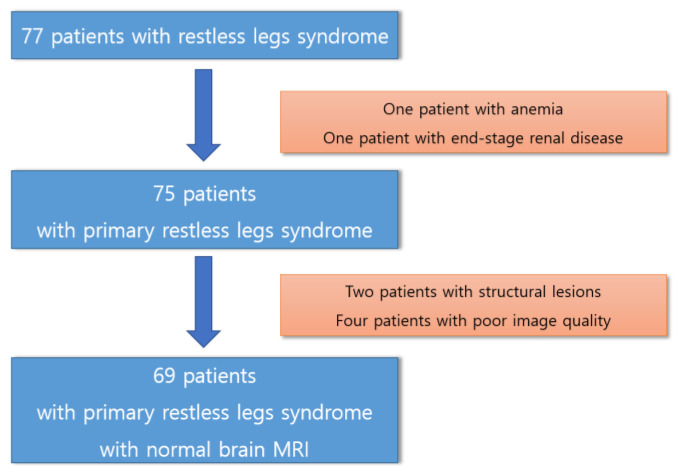
The selection process of the participants for this study.

**Figure 2 brainsci-13-01560-f002:**
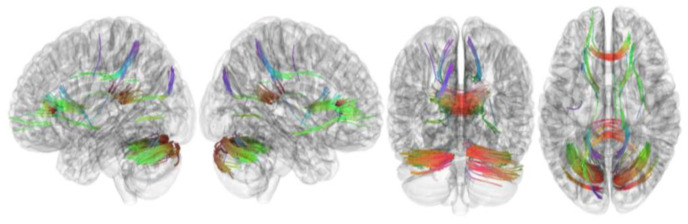
Tracks with FA positively correlated with RLS severity. Connectometry analysis revealed that the left cerebellum, right cerebellum, corpus callosum forceps minor, corpus callosum forceps major, corpus callosum body, and right cingulum frontoparietal tract showed FA positively correlated with RLS severity (FDR = 0.01). FA, fractional anisotropy; RLS, restless legs syndrome.

**Figure 3 brainsci-13-01560-f003:**
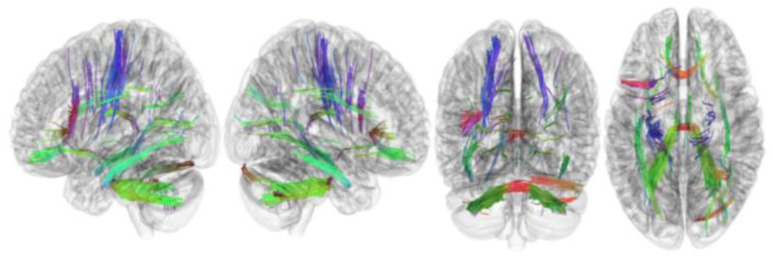
Tracks with FA negatively correlated with RLS severity. Connectometry analysis revealed that the middle cerebellar peduncle, left inferior longitudinal fasciculus, left corticospinal tract, corpus callosum forceps minor, right cerebellum, left frontal aslant tract, left dentatorubrothalamic tract, right inferior longitudinal fasciculus, left corticostriatal tract superior, and left cingulum parahippocamparietal tract showed FA negatively correlated with RLS severity (FDR = 0.04). FA, fractional anisotropy; RLS, restless legs syndrome.

**Figure 4 brainsci-13-01560-f004:**
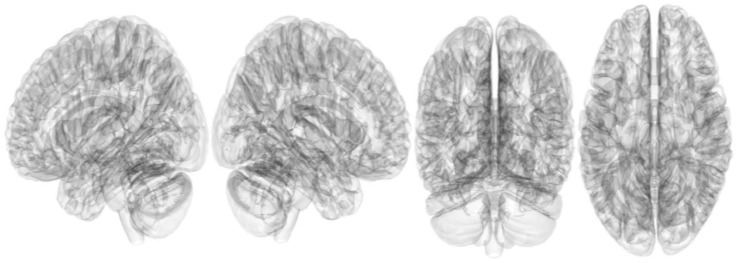
Tracks with QA positively correlated with RLS severity (FDR ≤ 0.05). The connectometry analysis find no significant result in tracks with QA positively correlated with RLS severity (FDR ≤ 0.05). QA, quantitative anisotropy; RLS, restless legs syndrome.

**Figure 5 brainsci-13-01560-f005:**
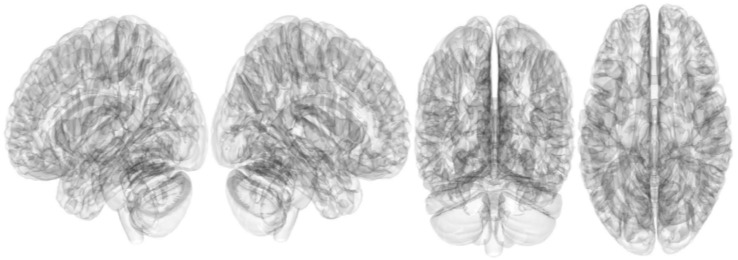
Tracks with QA negatively correlated with RLS severity. The connectometry analysis find no significant result in tracks with QA negatively correlated with RLS severity (FDR ≤ 0.05). QA, quantitative anisotropy; RLS, restless legs syndrome.

**Table 1 brainsci-13-01560-t001:** Demographic and clinical characteristics in patients with restless legs syndrome.

	Patients with Restless Legs Syndrome (N = 69)
Age, years	57.0 ± 6.6
Male, *n* (%)	20 (28.9)
Age of onset, years	47 (41.7–54.0)
Symptom duration, months	120 (39–162)
IRLS	27.1 ± 6.5
Disease-specific quality of life	8.7 ± 3.3
PSQI	12 (9.0–14.2)
ISI	16 (11–23)
HAS	7 (4–9)
HDS	8 (5–11)

HAS, Hospital Anxiety Scale; HDS, Hospital Depression Scale; IRLS, International Restless Legs Syndrome Severity Scale; ISI, Insomnia Severity Index; PSQI, Pittsburgh Sleep Quality Index.

**Table 2 brainsci-13-01560-t002:** The results of the connectometry analysis.

FA	
Tracks with positive correlation with RLS severity	Cerebellum, corpus callosum forceps minor, corpus callosum forceps major, corpus callosum body, and cingulum frontoparietal track
Tracks with negative correlation with RLS severity	Middle cerebellar peduncle, inferior longitudinal fasciculus, corticospinal tract, corpus callosum forceps minor, cerebellum, frontal aslant tract, dentato-rubrothalamic tract, inferior longitudinal fasciculus, corticostriatal tract superior, and cingulum parahippocampoparietal tract
QA	
Tracks with positive or negative correlation with RLS severity	None

FA, fractional anisotropy; QA, quantitative anisotropy.

## Data Availability

The data presented in this study are available on request from the corresponding author.

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
