# Peer review of "Correlation of Diffusion Tensor Tractography with Restless Legs Syndrome Severity"

_brainsci, 2023, doi:10.3390/brainsci13111560_

Round 1

Reviewer 1 Report

Comments and Suggestions for Authors

This prospective study investigated white matter tracts associated with restless legs syndrome (RLS) severity in 69 patients with primary RLS using correlational tractography based on diffusion tensor imaging. Fractional anisotropy (FA) and quantitative anisotropy (QA) were analyzed separately to understand white matter abnormalities in RLS patients. The study’s findings suggest that changes in white matter observed in RLS patients may be attributed to brain edema rather than structural damage to white matter tracts. Thus, this study highlights the importance of considering white matter alterations in understanding the pathophysiology of RLS and in developing effective treatment strategies. Overall, this study is interesting and the manuscript is well-written. I just have several minor comments.

1.     Please add some detail data in the abstract.

2.     Please describe in detail how to include patients.

3.     Please add one figure to show the selection of the patients based on the inclusion and exclusion criteria in this prospective study.

Author Response

1. Please add some detail data in the abstract.

: We followed your recommendation and described the results in more detail in the abstract.

2. Please describe in detail how to include patients.

: We added it in the method section. 

3. Please add one figure to show the selection of the patients based on the inclusion and exclusion criteria in this prospective study. 

: We made figure 1 as your recommendation.

Thank you for your wonderful review.

Reviewer 2 Report

Comments and Suggestions for Authors

The manuscript focused on a meaningful topic and was generally written well. Below I have some concerns and suggestions to improve the manuscript.

Firstly, it is noteworthy that no healthy controls were included in the present study. It should be considered as an limitation to be mentioned in my opinion.

In the present study, significant correlations between FA and RLS severity were found within a number of white matter tracts. However, how did the authors define these white matter tracts? Maybe it was based on an Atlas but I did't see any descriptions.

The authors used deterministic tractography in this study; however, probabilistic tractography has been suggested to be more accurate than deterministic tractography in recent years. This might be considered as a limitation and some recent studies using deterministic tractography should be cited, such as: https://www.frontiersin.org/articles/10.3389/fpsyt.2018.00391/full.

The following paragraph should be deleted in lines 185-186: "This section may be divided by subheadings. It should provide a concise and precise description of the experimental results, their interpretation, as well as the experimental conclusions that can be drawn."

There seems to be some errors in the reference list (e.g., the title of Ref No.35 is incomplete).

Author Response

1. Firstly, it is noteworthy that no healthy controls were included in the present study. It should be considered as a limitation to be mentioned in my opinion.

: We agree with you. We added it as a limitation for this study.

2. In the present study, significant correlations between FA and RLS severity were found within a number of white matter tracts. However, how did the authors define these white matter tracts? Maybe it was based on an Atlas but I did't see any descriptions.

: We used the ICBM152 adult template to discover white matter tracks. We added it in the method section.

3. The authors used deterministic tractography in this study; however, probabilistic tractography has been suggested to be more accurate than deterministic tractography in recent years. This might be considered a limitation and some recent studies using deterministic tractography should be cited, such as https://www.frontiersin.org/articles/10.3389/fpsyt.2018.00391/full.

: We agree with your opinion. We added it another limitation for this study in the discussion section. We additionally added the reference as your recommendation.

4. The following paragraph should be deleted in lines 185-186: "This section may be divided by subheadings. It should provide a concise and precise description of the experimental results, their interpretation, as well as the experimental conclusions that can be drawn."

: We were sorry about it. We deleted it.

5. There seem to be some errors in the reference list (e.g., the title of Ref No.35 is incomplete).

: We were sorry about it. We corrected it.

Thank you for your wonderful review.

Reviewer 3 Report

Comments and Suggestions for Authors

The paper describe an original work about association of restless leg syndrome and tractography MRI findings. This is the first report in the literature on this topic. The work is prospective and cross-sectional and the conclusion are very interesting as authors suggest the hypothesis that RLS is non related to structural changes.

Introduction is a little too long and I suggest should be shortened and synthesized.

Results are well exposed. I suggest the introduction of a table summarizing results. A comparison between symptoms onset time and FA QA results should be considered. Figures are good such as descriptions.

 Discussion is again a little too long. I suggest the reduce it with more focalization on results and less speculative aspects.

Comments on the Quality of English Language

Quality of writing is good and sentences are easily understandable

Author Response

1. Introduction is a little too long and I suggest should be shortened and synthesized.

: We reduced the introduction part as your recommendation. However, please understand that this journal requires a minimum of 4000 words.

2. Results are well exposed. I suggest the introduction of a table summarizing results.

: We added the table 2 as your recommendation.

3. Discussion is again a little too long. I suggest the reduce it with more focalization on results and less speculative aspects.

: We reduced the discussion as your recommendation.

Thank you for your wonderful review.

Reviewer 4 Report

Comments and Suggestions for Authors

This study is interesting and significant. Author speculated the change of FA may be come from brain edema, however this speculation was not supported with other findings or literatures. In other discussion, author explained the neural change in RLS, so these discussions did not support the theory of brain edema.

Author Response

1. This study is interesting and significant. Author speculated the change of FA may be come from brain edema, however this speculation was not supported with other findings or literatures. In other discussion, author explained the neural change in RLS, so these discussions did not support the theory of brain edema.

: When there is no change of QA with a significant change of FA in the white matter based on the DTI analysis, we could suspect this result originated from brain edema in the regions of the specific white matter tracts. However, we agree with you that this was simply a suspicion for this study. Thus, we changed the conclusion to your recommendation.

Thank you for your wonderful review.